# Online Learning Meets Adam: The Road of Interpretable Adaptive Optimizer Design

## Abstract

This paper explores the theoretical foundations of Adam, a widely used adaptive optimizer. Building on recent developments in non-convex optimization and online learning, particularly the discounted-to-nonconvex conversion framework, we present two aspects of results: First, we introduce clip-free FTRL, a novel variant of the classical Follow-the-Regularized-Leader (FTRL) algorithm. Unlike scale-free FTRL and the recently proposed $\beta$-FTRL, our clip-free variant eliminates the need for clipping operations, aligning more closely with Adam's practical implementation. This modification provides deeper theoretical insights into Adam's empirical success and aligns the theoretical framework with practical implementations. By incorporating a refined analysis, our second result establishes a theoretical guarantee for the Last Iterate Convergence (LIC) under the proposed discounts-to-nonconvex conversion algorithm in LIC, which differs from the previous guarantee that has convergence evenly distributed in all iterations. Additionally, we extend this result to provide the last iterate convergence guarantee for the popular $\beta$-FTRL algorithm under the same framework. However, the derived last iterate convergence of $\beta$-FTRL reveals a persistent fixed error, potentially suggesting either limitations in popular online learning methods or the need for additional assumptions about the objective function.

## 1 Introduction

Adaptive optimizers, particularly Adam (and AdamW)(Kingma, 2014; Loshchilov, 2017), are fundamental to the success of large-scale first-order optimization tasks, such as training large language models (Devlin, 2018; Radford et al., 2019; Bommasani et al., 2021; Touvron et al., 2023; Team et al., 2023). However, the theoretical underpinnings of Adam's performance remain elusive. While many efforts have been made to establish Adam's convergence equivalence to stochastic gradient descent (SGD), Adam usually demonstrates superior performance over SGD in practical scenarios. Unfortunately, most existing theoretical analyses are often not enlightening and fail to adequately account for Adam's key components, such as momentum update and bias correction for the first and second moments. These components are often seen as obstacles in theoretical proofs or are entirely disregarded (Li et al., 2024; Wang et al., 2024). Understanding the design principles behind Adam and explaining its performance advantages is an area in need of further exploration.

Recent advancements in the discounted-to-nonconvex conversion framework offer a promising avenue for understanding Adam's effectiveness. Cutkosky et al. (2023) introduced the online-to-nonconvex conversion framework, which deeply bridges non-smooth non-convex optimization with online learning. Building upon this foundation, subsequent works (Zhang & Cutkosky, 2024; Ahn & Cutkosky, 2024; Ahn et al., 2024) introduced the discounted-to-nonconvex conversion framework, offering new insights into the relationship between adaptive optimizers and online learners. This framework holds promise in revealing Adam's underlying mechanisms and effectiveness from a novel perspective.

The discounted-to-nonconvex framework consists of two primary components: the discounted-to-nonconvex conversion algorithm and the corresponding online learning algorithm. The theoretical foundation of the conversion algorithm ties the optimality condition of non-convex optimization, such as the gradient norm, to the discounted regret of the associated online learner. Notably, it commits the fact that each online learner is one-to-one corresponding to a specific optimizer, implying

that the design of an effective non-convex optimizer is tantamount to creating an online learner that minimizes discounted regret.

To further motivate this framework, the pioneering work (Ahn & Cutkosky, 2024) also proposes the performance criterion, gradient adaptivity, to demonstrate Adam's performance superiority over SGD. For instance, the online learning method $\beta$-FTRL (Orabona & Pál, 2018; Zhang et al., 2024), which is closely aligned with Adam, has been shown to better adapt to problem-dependent properties, i.e., offering stronger theoretical guarantees in scenarios where these properties are unknown.

In this paper, we extend the exploration of the discounted-to-nonconvex conversion framework, mitigating the discrepancy between the framework and the practical practices of Adam and obtaining notable theoretical improvements. Our key contributions are summarized as follows.

- We develop an online learning method, clip-free FTRL, which improves from previous methods by eliminating the unrealistic clipping operation present in the previous approaches. To achieve clip-free, we adopt components from Adam's update and incorporate them in the recently proposed $\beta$-FTRL, leading to close alignment with Adam. This results in achieving comparable discounted regret to $\beta$-FTRL without the need for clipping, offering a more comprehensive and practical understanding of Adam's performance.

- Recognizing the limitations of previous discounted-to-nonconvex conversion algorithm (Ahn & Cutkosky, 2024), which relied on Exponential Moving Average (EMA) parameters (Polyak & Juditsky, 1992; Ruppert, 1988) and spread convergence evenly across all iterations, we propose a new conversion algorithm. This algorithm establishes a theoretical guarantee that bridges discounted regret and last iterate guarantees in non-convex optimization. Additionally, we extend this framework to provide a last iterate convergence guarantee for the popular $\beta$-FTRL algorithm, and the results reveal the necessity of further investigation along this avenue.

## 1.1 RELATED WORK

Significant efforts have been dedicated to understanding Adam's superior performance from two perspectives: *convergence rate* and *adaptivity*. Various studies have analyzed Adam's convergence behavior, demonstrating that it achieves a convergence rate comparable to SGD for convex or smooth nonconvex functions under different stochastic gradient conditions and hyper-parameter configurations (Reddi et al., 2019; Zhou et al., 2018; Alacaoglu et al., 2020; Guo et al., 2021; Zhang et al., 2022; Wang et al., 2024). However, these analyses often fail to capture the contributions of Adam's core components. Moreover, it is well established that under these function assumptions, SGD already achieves the minimax optimal convergence rate. Beyond convergence speed, studying the adaptivity of Adam over complex deep-learning environments is also a popular trend to support the success of Adam. Wang et al. (2023) showed that AdaGrad, a precursor to Adam, can adapt to functions satisfying the generalized smoothness condition (Zhang et al., 2019), while plain SGD may converge arbitrarily slowly. Subsequent work (Li et al., 2024) extended this analysis to Adam, demonstrating its convergence under the generalized smoothness condition. Additionally, Crawshaw et al. (2022) highlighted the theoretical benefits of momentum updates, a component shared by Adam, for SignSGD algorithm under the generalized smoothness condition.

Another important line of inquiry is the Last Iterate Convergence (LIC), which has garnered substantial attention in the literature and has been widely utilized. Most existing works focus on characterizing the convergence behavior of SGD and SGD with Momentum (SGDM) under the convex and strongly convex assumptions (Ghadimi & Lan, 2012; Sebbouh et al., 2021; Jain et al., 2019; Tao et al., 2021). More recent work (Jin et al., 2022; Li et al., 2022) have extended these analyses to non-convex functions.

## 2 PRELIMINARIES

In this section, we introduce the necessary assumptions regarding the function, stochastic gradient, and domain, which are adopted from previous works (Cutkosky et al., 2023; Ahn & Cutkosky, 2024; Zhang & Cutkosky, 2024). Particularly, Assumption 2.1 and Assumption 2.2 . These assumptions are sufficient to design algorithms that achieve $(\lambda, \epsilon)$-stationary points, defined in Definition 2.3,

which is a common notation of optimality for non-convex and non-smooth optimization (Zhang & Cutkosky, 2024; Ahn & Cutkosky, 2024; Zhang et al., 2019; Jordan et al., 2023; Tian et al., 2022). It is worth noting that $(\lambda, \epsilon)$-stationary point is a relaxed version of Goldstein stationary point (Goldstein, 1977), but retains the desirable properties, supporting the conversion from stationarity for non-convex and non-smooth functions to first-order stationary points when the objective function is smooth.

**Assumption 2.1.** *Let $F : \mathbb{R}^d \to \mathbb{R}$ be a differentiable function with the following properties:*

- *The **function** $F$ is bounded below by $\inf_{\mathbf{x}} F(\mathbf{x})$. Meanwhile, defining $\Delta := F(\mathbf{x}_0) - \inf_{\mathbf{x}} F(\mathbf{x})$.*

- *The **function** $F$ is well-behaved, i.e., $\forall \mathbf{x}$ and $\mathbf{y}$, $F(\mathbf{x}) - F(\mathbf{y}) = \int_0^1 \langle \nabla F(\mathbf{x} + t(\mathbf{y} - \mathbf{x}), \mathbf{y} - \mathbf{x})\rangle dt$.*

- *The **function** $F$ is $G$-Lipshitz, i.e., $\forall \mathbf{x}, ||\nabla F(\mathbf{x})|| \leq G$.*

- *The **stochastic gradient** $\mathbf{g} \leftarrow StoGrad(\mathbf{x}, r)$ for randomness $r \in \mathcal{Z}$, and $\forall \mathbf{x}$ satisfies $\mathbb{E}[\mathbf{g}] = \nabla F(\mathbf{x})$ and $\mathbb{E}[||\mathbf{g} - \nabla F(\mathbf{x})||^2] \leq \sigma^2$. Note a quick result $\mathbb{E}[||\mathbf{g}||^2] \leq G^2 + \sigma^2$.*

**Assumption 2.2.** *Let domain $\mathcal{D} \subseteq \mathbb{R}^d$ be bounded, i.e., $\forall \mathbf{x} \in \mathcal{D}, ||\mathbf{x}|| \leq D$.*

**Definition 2.3** ($\lambda, \epsilon$-stationary point). *Supposing $F(\cdot) : \mathbb{R}^d \to \mathbb{R}$ is differentiable. Then $\mathbf{x}$ is a $(\lambda, \epsilon)$-stationary point of $F$ is $||\nabla F(\mathbf{x})||^{[\lambda]} \leq \epsilon$ where*

$$||\nabla F(\mathbf{x})||^{[\lambda]} := \inf_{p \in \mathcal{P}(\mathbb{R}^d), \mathbb{E}_{\mathbf{y} \sim p}[\mathbf{y}] = \mathbf{x}} \left\{ ||\mathbb{E}[\nabla F(\mathbf{y})]|| + \lambda \mathbb{E}[||\mathbf{y} - \mathbf{x}||^2] \right\}.$$

Additionally, we introduce the basics of online learning and key regret definitions, especially $\beta$-discounted regret, which are essential to our analysis. Online Linear Optimization (OLO) is an iterative algorithm: at each iteration $t$, the online learner selects an action and then receives a linear loss $\ell_t(\cdot) := \langle \mathbf{v}_t, \cdot \rangle$. The objective is to minimize the regret defined as the cumulative difference between the learner's loss and that of arbitrary comparator $\mathbf{u}$. Iterative optimization algorithms share a strong connection with adversarial online learning; for further details, we refer readers to Orabona (2019).

**Definition 2.4** (Static regret and $\beta$-discounted regret). *For a comparator $\mathbf{u}$, the regret is defined as $Regret_t(\mathbf{u}) := \sum_{s=1}^t (\ell_s(\mathbf{z}_s) - \ell_s(\mathbf{u}))$, where $\ell_t(\cdot) := \langle \mathbf{v}_t, \cdot \rangle$ in this work. $\beta$-**discounted regret** Further, supposing an algorithm discounting the loss by $\beta^{-s}$, i.e., $\ell_t^{[\beta]}(\cdot) = \beta^{-t}\ell_t(\cdot)$, the corresponding $\beta$-discounted regret is defined as $Regret_t^{[\beta]}(\mathbf{u}) := \beta^t \sum_{s=1}^t \left( \ell_s^{[\beta]}(\mathbf{z}_s) - \ell_s^{[\beta]}(\mathbf{u}) \right) = \sum_{s=1}^t \langle \beta^{t-s}\mathbf{v}_s, \mathbf{z}_s - \mathbf{u} \rangle$.*

# 3 BACKGROUND: BASICS OF DISCOUNTED-TO-NONCONVEX CONVERSION AND FTRL ALGORITHMS

## 3.1 DISCOUNTED-TO-NONCONVEX CONVERSION ALGORITHM

This work is built upon the discounted-to-nonconvex conversion developed in Ahn et al. (2024); Zhang & Cutkosky (2024); Ahn & Cutkosky (2024), outlined in Algorithm 1. Specifically, Ahn & Cutkosky (2024) provides the enlightening theoretical result of the conversion framework that the averaged gradient norm is upper bounded by the $\beta$-discounted regret of the associated online learner. The result commits the fact that designing an online learner that achieves a low discounted regret leads to an effective nonconvex optimizer, which again shines the one-to-one correspondence between online learners and optimizers.

To facilitate comparisons, we embed the proposed discounted-to-nonconvex conversion algorithm in Last Iterate Convergence (LIC) into the algorithm table here, i.e., Algorithm 1 in LIC. Additional details are provided in Section 5.

---

**Algorithm 1** Discounted-to-nonconvex conversion algorithm  in Last Iterate Convergence (LIC)

---

**Input:** Initial point $\mathbf{x}_0$, $T$, online learning algorithm $\mathcal{A}$ outputting $\mathbf{z}$, and discounting factor $\beta \in (0, 1)$

**for** $t = 1$ **to** $T$ **do**

    Receive $\mathbf{z}_t$ from $\mathcal{A}$

    Update $\mathbf{x}_t \leftarrow \mathbf{x}_{t-1} + \rho_t \mathbf{z}_t$, where $\rho_t \sim \text{Exp}(1)$ i.i.d.

    Compute $\mathbf{g}_t \leftarrow \text{StoGrad}(\mathbf{x}_t, \mathbf{r}_t)$ with freshly sampled randomness $\mathbf{r}_t$

    Send $\ell_t^{[\beta]}(\mathbf{z}_t) := \langle \beta^{T-t} \mathbf{g}_t, \mathbf{z}_t \rangle$ to $\mathcal{A}$

    Vanilla algorithm: $\bar{\mathbf{x}}_t \leftarrow \frac{\beta - \beta^t}{1 - \beta^t} \bar{\mathbf{x}}_{t-1} + \frac{1-\beta}{1-\beta^t} \mathbf{x}_t$;  LIC:  pass. {For output.}

**end for**

Return $\mathbf{x}_{\text{output}}$ where Vanilla algorithm: $\mathbf{x}_{\text{output}} \sim \text{Unif}(\bar{\mathbf{x}}_t : t \in [T])$;  LIC:  $\mathbf{x}_{\text{output}} = \mathbf{x}_T$.

---

## 3.2 Online learning algorithm: scale-free FTRL and $\beta$-FTRL

Scale-free Follow-the-Regularized-Leader algorithm (FTRL) is a well-known algorithm in online learning (Orabona & Pál, 2018). Under the online linear optimization setting, the goal of the algorithm is to choose the increment $\mathbf{z}_t$ at each iteration $t$ to minimize the regret $\sum_{t=1}^{T} \langle \mathbf{v}_t, \mathbf{z}_t - \mathbf{u} \rangle$. In terms of the increment selecting strategy, scale-free FTRL tracks the history of linear loss function $\ell_t(\cdot) = \langle \mathbf{v}_t, \cdot \rangle$ and adjusts its prediction based on accumulating of past $\mathbf{v}_t$ and on the selected regularizers $\{\frac{1}{2\alpha_t} || \cdot ||^2\}$, effectively leveraging all past information to refine future predictions. The scale-free FTRL is presented in Algorithm 2 (scale-free FTRL).

Ahn et al. (2024); Ahn & Cutkosky (2024) provide the key insight that incorporating the $\beta$-discounted regret into the scale-free FTRL algorithm, i.e., Algorithm 2 ($\beta$-FTRL), almost recovers the Adam algorithm, where the increment is choosing as a clipped version of $-\eta \frac{\sum_{s=1}^{t-1} \beta^{-s} \mathbf{v}_s}{\sqrt{\sum_{s=1}^{t-1} ||\beta^{-s} \mathbf{v}_s||^2}}$ originally. But it can be converted to Adam update (in vector form) $-\eta \frac{(1-\beta)/(1-\beta^t) \sum_{s=1}^{t-1} \beta^{t-s} \mathbf{v}_s}{\sqrt{(1-\beta_2)/(1-\beta_2^t) \sum_{s=1}^{t-1} \beta_2^{t-s} ||\mathbf{v}_s||^2}}$ by selecting $\beta_2$ as $\beta^2$, absorbing constant $(1-\beta)/(\sqrt{1-\beta_2})$ into $\eta$, adding coefficient $\beta^t/\sqrt{\beta_2^t}$ and bias correction $\sqrt{(1-\beta_2^t)}/(1-\beta^t)$, and omitting $\epsilon$.

To further motivate this framework, Theorem A.1 of Ahn & Cutkosky (2024) provides the static regret of scale-free FTRL. Regarding $\beta$-FTRL, it is convenient to derive the guarantee of the $\beta$-discounted regret as $\text{Regret}_t^{[\beta]}(\mathbf{u}) \leq \frac{4D\sqrt{1-\beta^T}(G+\sigma)}{\sqrt{1-\beta}}$ by substituting $\mathbf{v}_t$ with $\beta^{-t} \mathbf{g}_t$, demonstrated in Theorem 9 of Ahn & Cutkosky (2024).

Equipped with the discounted regret bound and the theoretical guarantee of the discounted-to-nonconvex conversion algorithm, we can conveniently derive the optimization guarantee in terms of the nonconvex optimization.

---

**Algorithm 2** Scale-free FTRL, $\beta$-FTRL

---

**Input:** Regularizers $\{\frac{1}{2\alpha_t} || \cdot ||^2\} : \mathbb{R}^d \to \mathbb{R}$, the bounded domain $\mathcal{D}$

**for** $t = 1$ **to** $T$ **do**

  •   Scale-free FTRL: $\mathbf{z}_t = \underset{\mathbf{z} \in \mathcal{D}}{\arg\min} \left[ \frac{1}{2\alpha_t} ||\mathbf{z}||^2 + \sum_{s=1}^{t-1} \langle \mathbf{v}_s, \mathbf{z} \rangle \right] = -\text{clip}_D \left( \eta \frac{\sum_{s=1}^{t-1} \mathbf{v}_s}{\sqrt{\sum_{s=1}^{t-1} ||\mathbf{v}_s||^2}} \right)^{[a]}$

  •   $\beta$-FTRL: $\mathbf{z}_t = \underset{\mathbf{z} \in \mathcal{D}}{\arg\min} \left[ \frac{1}{2\alpha_t} ||\mathbf{z}||^2 + \sum_{s=1}^{t-1} \langle \beta^{-s} \mathbf{v}_s, \mathbf{z} \rangle \right] = -\text{clip}_D \left( \eta \frac{\sum_{s=1}^{t-1} \beta^{-s} \mathbf{v}_s}{\sqrt{\sum_{s=1}^{t-1} ||\beta^{-s} \mathbf{v}_s||^2}} \right)^{[b]}$

    Receive $\ell_t(\cdot) = \langle \mathbf{v}_t, \cdot \rangle$

**end for**

[a] By selecting $\alpha_t$ as $\eta/\sqrt{\sum_{s=1}^{t-1} ||\mathbf{v}_s||^2}$. And $\text{clip}_D(\mathbf{x}) := \mathbf{x} \min(D/||\mathbf{x}||, 1)$.

[b] By selecting $\alpha_t$ as $\eta/\sqrt{\sum_{s=1}^{t-1} ||\beta^{-s} \mathbf{v}_s||^2}$.

---

## 4 CLIP-FREE FTRL

Before formally presenting our proposed methods, we first introduce the underlying intuition. Considering the iterative update game: $\mathbf{y}_t = \mathbf{y}_{t-1} + \mathbf{z}_t$ where $t \in \{1, \cdots, T\}$ and $\mathbf{z}_t := -\mathbf{g}_t$. Additionally, $\{\mathbf{y}_0, \mathbf{g}_t\}$ are bounded such that $\{||\mathbf{y}_0||, ||\mathbf{g}_t||\} \leq D$. The objective is to bound the squared norm of the output $\mathbf{y}_T$, denoted as $L := ||\mathbf{y}_T||^2$, by $\mathcal{O}(D^2)$, which leads to the formulation of *Initial Bound* below. However, the dependence on $T$ is undesirable, motivating us to remove this dependence in a subsequent formulation. Additionally, dependence on $\beta$ can also be problematic, especially when $\beta$ is very small (e.g., $\beta = 1 - \frac{1}{T}$). Eliminating this dependence yields a third formulation.

1. Initial Bound: $L = ||\mathbf{y}_0 - \sum_{t=1}^T \mathbf{g}_t||^2 \leq 2||\mathbf{y}_0||^2 + 2||\sum_{t=1}^T \mathbf{g}_t||^2 \leq 2(T+1)D^2$

2. Removing Dependence on $T$ by letting $\mathbf{z}_t = \beta^{T-t}\mathbf{g}_t$ where $\beta \in (0,1)$:

   • $L = ||\mathbf{y}_0 - \sum_t^T \beta^{T-t}\mathbf{g}_t||^2 \leq 2||\mathbf{y}_0||^2 + 2||\sum_t^T \beta^{T-t}\mathbf{g}_t||^2 \leq 2D^2 + \frac{2D^2}{(1-\beta)^2}$,

3. Removing Dependence on $T$ and $\beta$ by letting $\mathbf{z}_t = (1-\beta)\beta^{T-t}\mathbf{g}_t$ where $\beta \in (0,1)$:

   • $L = ||\mathbf{y}_0 - (1-\beta)\sum_t^T \beta^{T-t}\mathbf{g}_t||^2 \leq 2||\mathbf{y}_0||^2 + 2(1-\beta)^2||\sum_t^T \beta^{T-t}\mathbf{g}_t||^2 \leq 4D^2$.

This iterative framework emphasizes the importance of bounding the squared norm of the outputs, a principle that also applies to bounding the outputs of an online learner. Notably, both scale-free FTRL and $\beta$-FTRL in Algorithm 2 involve clipping operations to derive regret bounds, which serve as explicit bounding mechanisms. However, such clipping operations can be less reflective of real-world algorithm deployments. Moreover, reducing the squared norm of the online learner's outputs directly contributes to minimizing error in the variance term of the $(\lambda, \epsilon)$-stationarity, as shown in Lemma 10 of Ahn & Cutkosky (2024). This potentially results in a tighter non-convex optimization guarantee.

However, applying the combined strategy of discounting by $\beta^{T-t}$ and scaling by $1-\beta$ is not directly feasible for online learning methods like scale-free FTRL, which eliminates scaling due to its "scale-free" nature. Thus, adapted strategy for bounding the output of an online learner is required. In addition to bounding the outputs of online learners, it is crucial to maintain the same magnitude of $\beta$-discounted regret when developing a method for clip-free operation. To address both of these aspects, clip-free bounding implementation and consistent $\beta$-discounted regret, we now formally introduce our proposed method: clip-free FTRL, as detailed in Algorithm 3.

A key distinction between clip-free FTRL and other variants, such as scale-free FTRL or $\beta$-FTRL, is the removal of the clipping operation $\text{clip}_D(\cdot)$ within the increment. In contrast to $\beta$-FTRL, our method employs additional constant $(1-\beta)/(\sqrt{1-\beta_2})$ and coefficient $\beta^t/\sqrt{\beta_2^t}$ for increment $\mathbf{z}_t$. While these modifications may appear subtle, they contribute to achieving the theoretical advancements described in Section 4.1. Notably, clip-free FTRL almost recovers Adam's update, except for the bias correction terms.

---

**Algorithm 3** clip-free FTRL

---

**Input:** Regularizers $\{\frac{1}{2\alpha_t}|| \cdot ||^2\} : \mathbb{R}^d \to \mathbb{R}$

**for** $t = 1$ **to** $T$ **do**

$\quad \mathbf{z}_t = \arg\min\left[\frac{1}{2\alpha_t}||\mathbf{z}||^2 + (1-\beta)\sum_{s=1}^{t-1}\langle\beta^{t-1-s}\mathbf{v}_s, \mathbf{z}\rangle\right] = -\frac{\eta(1-\beta)\sum_{s=1}^{t-1}\beta^{t-1-s}\mathbf{v}_s}{\sqrt{(1-\beta_2)\sum_{s=1}^{t-1}\beta_2^{t-1-s}||\mathbf{v}_s||^2}}$ [a,b]

$\quad$ Receive $\ell_t(\cdot) = \langle\mathbf{v}_t, \cdot\rangle$

**end for**

---

[a] By selecting $\alpha_t$ as $\eta/\sqrt{(1-\beta_2)\sum_{s=1}^{t-1}\beta_2^{t-1-s}||\mathbf{v}_s||^2}$.

[b] Skipping update with zero loss: if $\mathbf{v}_t = \mathbf{0}$, freezing the updating of index $t$, i.e., omitting the zero term from subsequent summations and keeping the intermediate state at step $t+1$ identical to that at step $t$.

---

## 4.1 Discounted regret of clip-free FTRL

In this subsection, we aim to establish guarantees for clip-free FTRL in terms of the $\beta$-discounted regret. As highlighted in previous work (Ahn & Cutkosky, 2024; Ahn et al., 2024; Zhang & Cutkosky, 2024), a smaller discounted regret often leads to more effective non-convex optimizer.

To support the proof of the discounted regret guarantee for the proposed clip-free FTRL, Theorem 4.2, we first introduce Lemma 4.1, which characterizes the components of increment $\mathbf{z}_t$ in Algorithm 3. As shown in the result (C.2.) of the Lemma, $\mathbf{z}_t$ becomes independent of $T$ and $\beta$ when $\beta$ and $\beta_2$ are appropriately chosen. Results (C.1.) and (C.3.) serve as key steps in proving Theorem 4.2.

Finally, Theorem 4.2 presents the $\beta$-discounted regret for clip-free FTRL. Under Assumption 2.1, substituting $\mathbf{v}_t$ with $\mathbf{g}_t$ further gives the $\beta$-discount regret as $\text{Regret}_t^{[\beta]}(\mathbf{u}) \leq \frac{3D\sqrt{1-\beta_2^T}(G+\sigma)}{1-\beta}$. Compared to the $\beta$-discounted regret for scare-free FTRL, $\text{Regret}_t^{[\beta]}(\mathbf{u}) \leq \frac{4D\sqrt{1-\beta^T}(G+\sigma)}{\sqrt{1-\beta}}$, as presented in Theorem 9 of Ahn & Cutkosky (2024), the key distinction is that our method is clip-free.

Finally, to better motivate our algorithm design and results, we remark on the role of the additional discounting factors $\beta_2$, which differs from previous methods. At a high level, $\beta_2$ is specifically selected to ensure $\alpha_t$ in Algorithm 3 is a non-increasing sequence w.r.t. $t$. Furthermore, the relation between $\beta$ and $\beta_2$ is carefully designed to ensure $||\mathbf{z}_t||$ is bounded throughout the iterations.

**Lemma 4.1.** *Using the same notations in Algorithm 3. Further, defining (A.1).* $\tilde{\mathbf{v}}_{t,\beta,T} := (1-\beta)\beta^{T-t}\mathbf{v}_t$; *(A.2).* $\tilde{\tilde{\mathbf{v}}}_{t,\beta,T} := (1-\beta)\beta^{T-t}||\mathbf{v}_t||^2$, *we have following re-formulations, (B.1).* $\alpha_t = \frac{\eta}{\sqrt{\sum_{s=1}^{t-1}\tilde{\tilde{\mathbf{v}}}_{s,\beta_2,t-1}}}$; *(B.2).* $\mathbf{z}_t = -\frac{\eta \sum_{s=1}^{t-1}\tilde{\mathbf{v}}_{s,\beta,t-1}}{\sqrt{\sum_{s=1}^{t-1}\tilde{\tilde{\mathbf{v}}}_{s,\beta_2,t-1}}}$. *Further, assuming* $\beta_2 \in (1-\frac{1}{a(T-1)}, 1)$ *and* $\beta \in (\beta_2, \sqrt{\beta_2})$. *Meanwhile, $a$ is some tunable parameter satisfying* $\max_{s \in [t-1]}||\mathbf{v}_s||^2 \leq (a-1)||\mathbf{v}_t||^2$ *and* $a > 1$, *we have*

> *(C.1).* $\alpha_t$ *is a non-increasing sequence w.r.t. $t$;*
>
> *(C.2).* $||\mathbf{z}_t|| \leq \eta$. *I.e., the norm of increment is bounded;*
>
> *(C.3).* $||\tilde{\mathbf{v}}_{t,\beta,T}||^2 \leq (1-\beta_2)\tilde{\tilde{\mathbf{v}}}_{t,\beta_2,T}$.

**Theorem 4.2** (Discounted regret of clip-free FTRL). *Using the same notations and hyper-parameter selection of Lemma 4.1, for all $T > 0$, loss sequence $\tilde{\mathbf{v}}_{1,\beta,T}, \cdots, \tilde{\mathbf{v}}_{T,\beta,T}$, comparator $\mathbf{u} \in \mathcal{D}$, i.e., $||\mathbf{u}|| \leq D$ (Assumption 2.2). Clip-free FTRL guarantees the $\beta$-discounted regret bound of* $\text{Regret}_t^{[\beta]}(\mathbf{u}) \leq \frac{3D\sqrt{1-\beta_2}}{1-\beta}\sqrt{\sum_{t=1}^{T}\beta_2^{T-t}||\mathbf{v}_t||^2}$.

The proof of Lemma 4.1 is presented in Appendix A.1. The proof of Theorem 4.2 is inspired by techniques from Ahn & Cutkosky (2024); Ahn et al. (2024); Tim (2021) and is presented in Appendix A.1.

## 5 Last iterate convergence of adaptive nonconvex optimization

In contrast to the previous section, which focused on online learning methods, this section delves into the conversion algorithm. In Section 5.1, we introduce a new conversion algorithm and derive its theoretical guarantee in terms of discounted regret and last iterate guarantee of gradient norm, as stated in Theorem 5.1. This independent result serves as a critical connection between the discounted regret of online learning algorithms and the last-iterate guarantees of non-convex optimization. In the subsequent Section 5.2, we provide the last iterate convergence for $\beta$-FTRL in non-convex optimization, building upon the new conversion algorithm.

### 5.1 Guarantee of discounted-to-nonconvex conversion in LIC

In this subsection, we provide the guarantee of the proposed conversion algorithm (Algorithm 1 in LIC). Specifically, Algorithm 1 in LIC represents our new conversion algorithm, with the main differences from the vanilla conversion algorithm highlighted in gray. However, there is no great

burden: the intermediate status $\hat{\mathbf{x}}_t$ (for EMA) is removed, and the last iterate is selected as the output.

As shown in Theorem 5.1, the gradient norm is upper bounded by the $\beta$-discounted regret of the associated online learner, effectively bridging the nonconvex optimization with the design of online learning algorithms. Additionally, we observe the indication of the last iterate convergence within this framework, i.e., $\mathbb{P}(\mathbf{x}_t)$.

**Theorem 5.1.** *Supposing that $F$ satisfies Assumption 2.1. Then for the comparator sequence chosen as $\mathbf{u}_t := -D \frac{\sum_{s=1}^{t} \beta^{-s} \nabla F(\mathbf{x}_s)}{|| \sum_{s=1}^{t} \beta^{-s} \nabla F(\mathbf{x}_s) ||}$. The Algorithm 1 in LIC guarantees*

$$\mathbb{E}\left[ ||\mathbb{E}_{\mathbb{P}(\mathbf{x}_t)} \nabla F(\mathbf{x}_t)] \, ||\right] \leq \frac{(1-\beta)\Delta}{(1-\beta^T)\beta D} + \frac{2(1-\beta)^{\frac{3}{2}}(G+\sigma)T}{\sqrt{1-\beta^T \beta}} + \frac{2\sqrt{(1-\beta)}}{\sqrt{1-\beta^T}}\sigma \tag{1}$$

$$+ \frac{(1-\beta)^2 T}{(1-\beta^T)\beta D} \mathbb{E}\left[\mathbb{E}_{t\sim[T]} Regret_t^{[\beta]}(\mathbf{u}_t)\right] + \frac{1-\beta}{(1-\beta^T)D} \mathbb{E}\left[Regret_T^{[\beta]}(\mathbf{u}_T)\right]$$

*where $\mathbf{x}_t$ is distributed over $\{\mathbf{x}_t\}_{t=1}^T$ as $\mathbb{P}(\mathbf{x}_t) = \frac{(1-\beta)\beta^{T-t}}{1-\beta^T}$ for $t = 1, 2, \cdots, T$. The outer expectation $\mathbb{E}[\cdot]$ is w.r.t. randomness $\rho$ and stochastic gradient randomness $r$.*

The proof is inspired by techniques of Lemma 7 in Ahn & Cutkosky (2024) and is presented below.

*Proof.* . We start from

$$\sum_{t=1}^{T} (F(\mathbf{x}_t) - F(\mathbf{x}_{t-1})) = \sum_{t=1}^{T}(1 - \beta^{T-t+1}) (F(\mathbf{x}_t) - F(\mathbf{x}_{t-1})) + \sum_{t=1}^{T} \beta^{T-t+1} (F(\mathbf{x}_t) - F(\mathbf{x}_{t-1}))$$

$$= \sum_{n=1}^{T}\sum_{t=1}^{n} \beta^{n-t}(1 - \beta)(F(\mathbf{x}_t) - F(\mathbf{x}_{t-1})) + \sum_{t=1}^{T} \beta^{T-t+1} (F(\mathbf{x}_t) - F(\mathbf{x}_{t-1}))$$

$$-\Delta \leq \sum_{n=1}^{T}\sum_{t=1}^{n} \beta^{n-t}(1 - \beta)(F(\mathbf{x}_t) - F(\mathbf{x}_{t-1})) + \sum_{t=1}^{T} \beta^{T-t+1} (F(\mathbf{x}_t) - F(\mathbf{x}_{t-1}))$$

where the last inequality is by the fact that $-\sum_{t=1}^{T} ((F(\mathbf{x}_t) - F(\mathbf{x}_{t-1})) = F(\mathbf{x}_0) - F(\mathbf{x}_T) \leq F(\mathbf{x}_0) - \inf_{\mathbf{x}} F(\mathbf{x}) = \Delta$.

Taking expectation on both sizes w.r.t. randomness $\rho$ and stochastic gradient randomness $r$, meanwhile simplifying $\mathbb{E}_{\rho,r}[\cdot]$ as $\mathbb{E}[\cdot]$, we get

$$-\Delta \leq \underbrace{\mathbb{E}\left[\sum_{n=1}^{T}\sum_{t=1}^{n} \beta^{n-t}(1 - \beta) (F(\mathbf{x}_t) - F(\mathbf{x}_{t-1}))\right]}_{\text{Part A}} + \underbrace{\mathbb{E}\left[\sum_{t=1}^{T} \beta^{T-t+1} (F(\mathbf{x}_t) - F(\mathbf{x}_{t-1}))\right]}_{\text{Part B}}$$

- Part A can be decomposed as

$$\underbrace{\mathbb{E}\left[\sum_{n=1}^{T}\sum_{t=1}^{n} \beta^{n-t}(1 - \beta) (F(\mathbf{x}_t) - F(\mathbf{x}_{t-1}))\right]}_{\text{Part A}} \stackrel{(i)}{=} (1-\beta)\mathbb{E}\left[\sum_{n=1}^{T}\sum_{t=1}^{n} \beta^{n-t}\langle \nabla F(\mathbf{x}_t), \mathbf{z}_t\rangle\right]$$

$$= (1-\beta)\mathbb{E}\left[\sum_{n=1}^{T}\sum_{t=1}^{n} \beta^{n-t}\langle \mathbf{g}_t, \mathbf{z}_t\rangle\right] = (1-\beta)\mathbb{E}\left[\sum_{n=1}^{T}\sum_{t=1}^{n} \beta^{n-t} (\langle \mathbf{g}_t, \mathbf{z}_t - \mathbf{u}_n\rangle + \langle \mathbf{g}_t, \mathbf{u}_n\rangle)\right]$$

$$\leq (1-\beta)\mathbb{E}\left[\sum_{t=1}^{T} Regret_t^{[\beta]}(\mathbf{u}_t)\right] + (1-\beta)\mathbb{E}\left[\sum_{n=1}^{T} \sqrt{|| \sum_{t=1}^{n} \beta^{n-t}\mathbf{g}_t||^2 ||\mathbf{u}_n||^2}\right]$$

$$= (1-\beta)T\mathbb{E}\left[\sum_{t=1}^{T} \frac{1}{T}Regret_t^{[\beta]}(\mathbf{u}_t)\right] + (1-\beta)D(G+\sigma)\sum_{t=1}^{T} \sqrt{\frac{1-\beta^{2t}}{1-\beta}}$$

$$= (1-\beta)T\mathbb{E}\left[\mathbb{E}_{t\sim[T]}Regret_t^{[\beta]}(\mathbf{u}_t)\right] + \sqrt{(1-\beta)(1-\beta^T)}D(G+\sigma)T$$

where the last equality is probability conversion, and the (i) applies Lemma 3.1 in Zhang & Cutkosky (2024).

- Part B can be decomposed as

$$\underbrace{\mathbb{E}\left[\sum_{t=1}^T \beta^{T-t+1}\left(F(\mathbf{x}_t) - F(\mathbf{x}_{t-1})\right)\right]}_{\text{Part B}} \overset{(i)}{=} \mathbb{E}\left[\sum_{t=1}^T \beta^{T-t+1}\langle\nabla F(\mathbf{x}_t), \mathbf{z}_t\rangle\right]$$

$$= \mathbb{E}\left[\sum_{t=1}^T \beta^{T-t+1}\left(\langle\nabla F(\mathbf{x}_t), \mathbf{u}_T\rangle + \langle\nabla F(\mathbf{x}_t) - \mathbf{g}_t, \mathbf{z}_t - \mathbf{u}_T\rangle + \langle\mathbf{g}_t, \mathbf{z}_t - \mathbf{u}_T\rangle\right)\right]$$

$$= \mathbb{E}\left[\sum_{t=1}^T \beta^{T-t+1}\left(\underbrace{\langle\nabla F(\mathbf{x}_t), \mathbf{u}_T\rangle}_{\text{Part 1}} + \underbrace{\langle\nabla F(\mathbf{x}_t) - \mathbf{g}_t, -\mathbf{u}_T\rangle}_{\text{Part 2}} + \underbrace{\langle\mathbf{g}_t, \mathbf{z}_t - \mathbf{u}_T\rangle}_{\text{Part 3}}\right)\right].$$

Here the (i) applies Lemma 3.1 in Zhang & Cutkosky (2024), and the last equality is by the fact $\mathbb{E}_r\left[\langle\nabla F(\mathbf{x}_t) - \mathbf{g}_t, \mathbf{z}_t\rangle\right] = 0$.

- Part B.1 can be further re-formulated as

$$\mathbb{E}\left[\sum_{t=1}^T \beta^{T-t+1}\langle\nabla F(\mathbf{x}_t), \mathbf{u}_T\rangle\right]$$

$$= \beta\mathbb{E}\left[\left\langle\sum_{t=1}^T \beta^{T-t}\nabla F(\mathbf{x}_t), -D\frac{\sum_{t=1}^T \beta^{T-t}\nabla F(\mathbf{x}_t)}{||\sum_{t=1}^T \beta^{T-t}\nabla F(\mathbf{x}_t)||}\right\rangle\right]$$

$$= \beta\frac{1-\beta^T}{1-\beta}\mathbb{E}\left[\left\langle\sum_{t=1}^T \frac{1-\beta}{1-\beta^T}\beta^{T-t}\nabla F(\mathbf{x}_t), -D\frac{\sum_{t=1}^T \frac{1-\beta}{1-\beta^T}\beta^{T-t}\nabla F(\mathbf{x}_t)}{||\sum_{t=1}^T \frac{1-\beta}{1-\beta^T}\beta^{T-t}\nabla F(\mathbf{x}_t)||}\right\rangle\right]$$

$$= -\beta D\frac{1-\beta^T}{1-\beta}\mathbb{E}\left[||\mathbb{E}_{\mathbb{P}(\mathbf{x}_t)}\nabla F(\mathbf{x}_t)||\right]$$

where the last equality is probability conversion, where $\mathbf{x}_t$ is distributed over $\{\mathbf{x}_t\}_{t=1}^T$ as $\mathbb{P}(\mathbf{x}_t) = \frac{(1-\beta)\beta^{T-t}}{1-\beta^T}$ for $t = 1, 2, \cdots, T$.

- Part B.2 can be further re-formulated as

$$\mathbb{E}\left[\sum_{t=1}^T \beta^{T-t+1}\langle\nabla F(\mathbf{x}_t) - \mathbf{g}_t, -\mathbf{u}_T\rangle\right] \leq \beta\mathbb{E}\left[\sqrt{||\sum_{t=1}^T \beta^{T-t}\left(F(\mathbf{x}_t) - \mathbf{g}_t\right)||^2||\mathbf{u}_T||^2}\right]$$

$$\leq \beta D\mathbb{E}\left[\sqrt{\sum_{t=1}^T \beta^{2T-2t}||\left(F(\mathbf{x}_t) - \mathbf{g}_t\right)||^2}\right] \leq \sigma\beta D\sqrt{\frac{1-\beta^T}{1-\beta}}$$

where the first inequality is due to Triangle inequality, the last inequality is due to the bounded variance assumption on the stochastic gradient oracle.

- Part B.3 can be further re-formulated as

$$\mathbb{E}\left[\sum_{t=1}^T \beta^{T-t+1}\langle\mathbf{g}_t, \mathbf{z}_t - \mathbf{u}_T\rangle\right] = \beta\mathbb{E}\left[\text{Regret}_T^{[\beta]}(\mathbf{u}_T)\right].$$

Combining the final results of Part A and Part B, we have that

$$-\Delta \leq (1-\beta)T\mathbb{E}\left[\mathbb{E}_{t\sim[T]}\text{Regret}_t^{[\beta]}(\mathbf{u}_t)\right] + \sqrt{(1-\beta)(1-\beta^T)}D(G+\sigma)T$$

$$- \beta D\frac{1-\beta^T}{1-\beta}\mathbb{E}\left[||\mathbb{E}_{\hat{\mathbf{x}}}\nabla F(\hat{\mathbf{x}})|||\right] + \sigma\beta D\sqrt{\frac{1-\beta^T}{1-\beta}} + \beta\mathbb{E}\left[\text{Regret}_T^{[\beta]}(\mathbf{u}_T)\right]$$

$$\mathbb{E}\left[||\mathbb{E}_{\hat{\mathbf{x}}}\nabla F(\hat{\mathbf{x}})|||\right] \leq \frac{(1-\beta)\Delta}{(1-\beta^T)\beta D} + \frac{2(1-\beta)^{\frac{3}{2}}(G+\sigma)T}{\sqrt{1-\beta^T}\beta} + \frac{2\sqrt{(1-\beta)}}{\sqrt{1-\beta^T}}\sigma$$

$$+ \frac{(1-\beta)^2 T}{(1-\beta^T)\beta D}\mathbb{E}\left[\mathbb{E}_{t\sim[T]}\text{Regret}_t^{[\beta]}(\mathbf{u}_t)\right] + \frac{1-\beta}{(1-\beta^T)D}\mathbb{E}\left[\text{Regret}_T^{[\beta]}(\mathbf{u}_T)\right]$$

which concludes our proof. $\qquad\square$

## 5.2 LAST ITERATE GUARANTEE FOR $\beta$-FTRL IN NON-CONVEX OPTIMIZATION

In this subsection, we aim to establish the last iterate guarantees for nonconvex optimization in terms of the selected $(\lambda, \epsilon)$-stationarity under the conversion framework presented in the above section. To support the proof of our final result, Theorem 5.3, we first introduce the following Lemma 5.2, which characterizes the decay component, i.e., $\frac{1-\beta}{1-\beta^T} = \mathcal{O}(\frac{1}{T})$, in equation 1;

It is worth mentioning that our statement on the last-iterate guarantee offers the insight that the last iterate has a higher likelihood of being selected compared to other iterations, distinguishing it from the common last-iterate guarantee statements in Liu & Zhou (2023); Li et al. (2022).

**Lemma 5.2.** *Supposing $\beta = 1 - \frac{1}{T}$ and $T \geq 2$, we have $(1-\beta^T) > 0.632$ and $\beta > 0.5$.*

*Proof.* Consider the general form of an exponential limit $\lim_{t\to\infty}\left(1+\frac{a}{t}\right)^t = e^a$, which is monotonically increasing w.r.t. $t$. Thus, we have $(1-\beta^T) = 1 - (1-\frac{1}{T})^T \geq 1 - \lim_{t\to\infty}\left(1+\frac{-1}{t}\right)^t = 1 - e^{-1} \approx 0.632$. $\qquad\square$

Equipped with the above Lemma 5.2 and Theorem 5.1, the nonconvex optimization guarantee in terms of $(\lambda, \epsilon)$-stationarity is presented as Theorem 5.3. We observe that Algorithm 1 in LIC selecting $\mathcal{A}$ as $\beta$-FTRL converges to a region near $(\lambda, \epsilon)$-stationarity, where the error is bounded by $\mathcal{O}(\lambda + \Delta)$ and is independent over $G$ and $\sigma$.

**Theorem 5.3.** *Supposing $F$ satisfies Assumption 2.1 and consider $\forall \lambda > 0$. Algorithm 1 in LIC selecting $\mathcal{A}$ as $\beta$-FTRL and $\beta = 1 - \frac{1}{T}$ guarantees*

$$||\nabla F(\hat{\mathbf{x}})||^{[\lambda]} \leq \mathcal{O}(\lambda + \Delta) + \frac{24(G+\sigma)}{\sqrt{T}},$$

*where $\hat{\mathbf{x}}$ is distributed over $\{\mathbf{x}_t\}_{t=1}^T$ as $\mathbb{P}(\mathbf{x}_t) = \frac{(1-\beta)\beta^{T-t}}{1-\beta^T}$.*

*Proof.* Denote $\hat{\mathbf{x}} := \mathbb{E}_{\mathbb{P}(\mathbf{x}_t)}[\mathbf{x}_t]$ where $\mathbb{P}(\mathbf{x}_t) = \frac{(1-\beta)\beta^{T-t}}{1-\beta^T}$, then the optimality condition (Definition 2.3) gives

$$||\nabla F(\hat{\mathbf{x}})||^{[\lambda]} = \inf\left\{||\mathbb{E}_{\mathbb{P}(\mathbf{x}_t)}[\nabla F(\mathbf{x}_t)]|| + \lambda\mathbb{E}_{\mathbb{P}(\mathbf{x}_t)}[||\mathbf{x}_t - \hat{\mathbf{x}}||^2]\right\}.$$

**Fisrtly, we deal with $\lambda\mathbb{E}_{\mathbb{P}(\mathbf{x}_t)}[||\mathbf{x}_t - \hat{\mathbf{x}}||^2]$.**

By Lemma 10 of Ahn & Cutkosky (2024), we have $\mathbb{E}_{\mathbb{P}(\mathbf{x}_t)}[||\mathbf{x}_t - \hat{\mathbf{x}}||^2] \leq \frac{12D^2}{(1-\beta)^2}$.

**Secondly, we deal with $||\mathbb{E}_{p(\mathbf{x})}[\nabla F(\mathbf{x}_t)]||$.**

Given inequality 1 in Theorem 5.1, substituting $\mathbb{E}\left[\text{Regret}_t^{[\beta]}(\mathbf{u}_t)\right]$ with corresponding upper bound $\frac{\sqrt{1-\beta^T}4D(G+\sigma)}{\sqrt{1-\beta}}$ further re-formulated inequality 1 as

$$||\mathbb{E}_{\mathbb{P}(\mathbf{x}_t)}[\nabla F(\mathbf{x}_t)]|| \leq \frac{(1-\beta)\Delta}{(1-\beta^T)\beta D} + \frac{6(1-\beta)^{\frac{3}{2}}(G+\sigma)T}{\sqrt{1-\beta^T}\beta} + \frac{4\sqrt{1-\beta}(G+\sigma)}{\sqrt{1-\beta^T}} + \frac{\sqrt{(1-\beta)}}{\sqrt{1-\beta^T}}\sigma$$

Combining the above two results, we have

$$||\nabla F(\hat{\mathbf{x}})||^{[\lambda]} \leq \left( \frac{12\lambda D^2}{(1-\beta)^2} + \frac{(1-\beta)\Delta}{(1-\beta^T)\beta D} \right) + \frac{6(1-\beta)^{\frac{3}{2}}(G+\sigma)T}{\sqrt{1-\beta^T}\beta} + \frac{4\sqrt{1-\beta}(G+\sigma)}{\sqrt{1-\beta^T}} + \frac{\sqrt{(1-\beta)}}{\sqrt{1-\beta^T}}\sigma$$

Supposing $\beta := 1 - \frac{1}{T}, T > 2$ and considering Lemma 5.2, i.e, $1 - \beta^T > 0.632, \sqrt{1-\beta^T} > 0.794$, and $\beta > 0.5$, Meanwhile $D = \mathcal{O}(1-\beta)$ ,the above inequality can be further reformulated as

$$||\nabla F(\hat{\mathbf{x}})||^{[\lambda]} < \mathcal{O}(\lambda+\Delta) + \frac{16(G+\sigma)}{\sqrt{T}} + \frac{6(G+\sigma)}{\sqrt{T}} + \frac{2\sigma}{\sqrt{T}} < \mathcal{O}(\lambda+\Delta) + \frac{24(G+\sigma)}{\sqrt{T}},$$

which concludes the proof.

$\square$

## 6 DISCUSSION

In this work, we propose the clip-free FTRL algorithm, expanding on pivotal contributions within the discounted-to-online conversion framework, which is increasingly influential for analyzing and designing adaptive optimizers. The introduced modification to the plain $\beta$-FTRL is subtle yet impactful. Our analysis sheds light on the underlying mechanism of effective components of Adam. However, our findings come with limitations:

- Compared with the Adam update (in its vector form) $-\eta \frac{(1-\beta)/(1-\beta^t)\sum_{s=1}^{t-1}\beta^{t-s}\mathbf{v}_s}{\sqrt{(1-\beta_2)/(1-\beta_2^t)\sum_{s=1}^{t-1}\beta_2^{t-s}||\mathbf{v}_s||^2}}$, clip-free FTRL suggests an update of $-\eta \frac{(1-\beta)\sum_{s=1}^{t-1}\beta^{t-s}\mathbf{v}_s}{\sqrt{(1-\beta_2)\sum_{s=1}^{t-1}\beta_2^{t-s}||\mathbf{v}_s||^2}}$. The numerator $\mathbf{m}_{t-1} := (1-\beta)\sum_{s=1}^{t-1}\beta^{t-s}\mathbf{v}_s$ recovers the classical momentum update $\mathbf{m}_t = \beta\mathbf{m}_{t-1} + (1-\beta)\mathbf{v}_t$ . However, the discrepancy arising from the missing bias correction terms in clip-free FTRL remains unexplored.

- Additionally, while Lemma 4.1 characterizes the relationship between $\beta$ and $\beta_2$, the practical values typically used in applications, i.e., $\beta = 0.9$ and $\beta_2 = 0.999$, do not conform to the theoretical condition $\beta \in (\beta_2, \sqrt{\beta_2})$. To achieve the bounded increment $||\mathbf{z}_t||$ in the proof of Lemma 4.1, we sequentially apply the Triangle inequality and Cauchy-Schwarz inequality, referring to *Verifying (C.2)* in Appendix A.1, which may introduce larger errors and restrict the selection range for $\beta$. However, it holds the potential to achieve more relaxed conditions for $\beta$ and presents an avenue for future work.

Regarding our analysis on the last iterate convergence, we introduce a new conversion algorithm (Algorithm 1 in LIC) and provide a corresponding guarantee in Theorem 5.1. This result establishes a bridge between the last-iterate convergence in nonconvex optimization and the the $\beta$-discounted regret of online learning algorithms, which could be of independent interest. Nonetheless, a subsequent result built upon this conversion framework suggests an unsatisfactory convergence behaviors of popular $\beta$-FTRL, which necessitates further investigation.

## ETHICS STATEMENT

Our work primarily focuses on theoretical and practical developments in optimization methods, which potentially enable efficient model training of deep model optimization tasks. However, we are also aware that the advancements may have broader implications, some of which could potentially have negative social impacts, such as misuse of the method in malicious application developments.

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

## A    TECHNICAL PROOFS

### A.1    PROOFS FOR SECTION: DISCOUNTED REGRET OF CLIP-FREE FTRL

**Lemma 4.1.** *Using the same notations in Algorithm 3. Further, defining (A.1). $\tilde{\mathbf{v}}_{t,\beta,T} := (1 - \beta)\beta^{T-t}\mathbf{v}_t$; (A.2). $\tilde{\tilde{\mathbf{v}}}_{t,\beta,T} := (1 - \beta)\beta^{T-t}||\mathbf{v}_t||^2$, we have following re-formulations, (B.1). $\alpha_t = \frac{\eta}{\sqrt{\sum_{s=1}^{t-1}\tilde{\tilde{\mathbf{v}}}_{s,\beta_2,t-1}}}$; (B.2). $\mathbf{z}_t = -\frac{\eta\sum_{s=1}^{t-1}\tilde{\mathbf{v}}_{s,\beta,t-1}}{\sqrt{\sum_{s=1}^{t-1}\tilde{\tilde{\mathbf{v}}}_{s,\beta_2,t-1}}}$. Further, assuming $\beta_2 \in (1 - \frac{1}{a(T-1)}, 1)$ and $\beta \in (\beta_2, \sqrt{\beta_2})$. Meanwhile, $a$ is some tunable parameter satisfying $\max_{s\in[t-1]}||\mathbf{v}_s||^2 \le (a - 1)||\mathbf{v}_t||^2$ and $a > 1$, we have*

> *(C.1). $\alpha_t$ is a non-increasing sequence w.r.t. $t$;*
>
> *(C.2). $||\mathbf{z}_t|| \le \eta$. I.e., the norm of increment is bounded;*
>
> *(C.3). $||\tilde{\mathbf{v}}_{t,\beta,T}||^2 \le (1 - \beta_2)\tilde{\tilde{\mathbf{v}}}_{t,\beta_2,T}$.*

*Proof.* Defining (A.1). $\tilde{\mathbf{v}}_{t,\beta,T} := (1 - \beta)\beta^{T-t}\mathbf{v}_t$; (A.2). $\tilde{\tilde{\mathbf{v}}}_{t,\beta,T} := (1 - \beta)\beta^{T-t}||\mathbf{v}_t||^2$, then, we have the following formulations by the definitions

- (B.1). $\sum_{s=1}^{t-1}\tilde{\tilde{\mathbf{v}}}_{s,\beta_2,t-1} = (1 - \beta_2)\sum_{s=1}^{t-1}\beta_2^{t-1-s}||\mathbf{v}_s||^2$. Thus $\alpha_t = \frac{\eta}{\sqrt{\sum_{s=1}^{t-1}\tilde{\tilde{\mathbf{v}}}_{s,\beta_2,t-1}}}$;

- (B.2). $\sum_{s=1}^{t-1}\tilde{\mathbf{v}}_{s,\beta,t-1} = (1 - \beta)\sum_{s=1}^{t-1}\beta^{t-1-s}\mathbf{v}_s$. Thus, $\mathbf{z}_t = -\frac{\eta\sum_{s=1}^{t-1}\tilde{\mathbf{v}}_{s,\beta,t-1}}{\sqrt{\sum_{s=1}^{t-1}\tilde{\tilde{\mathbf{v}}}_{s,\beta_2,t-1}}}$;

- (B.3). $\sum_{s=1}^{t}\tilde{\tilde{\mathbf{v}}}_{s,\beta_2,t} = \beta_2\sum_{s=1}^{t-1}\tilde{\tilde{\mathbf{v}}}_{s,\beta_2,t-1} + (1 - \beta_2)||\mathbf{v}_t||^2$. Thus, the value within the square root of the denominator satisfies the convex combination type of update rule (classical momentum update).

**Verifying (C.1).** $\alpha_t \ge \alpha_{t+1}$

Assuming $\mathbf{v}_t = \mathbf{0}$, we have $\alpha_t = \alpha_{t+1}$ by the algorithm design.

Assuming $\mathbf{v}_t \ne \mathbf{0}$ and given (B.3). $\sum_{s=1}^{t}\tilde{\tilde{\mathbf{v}}}_{s,\beta_2,t} = \beta_2\sum_{s=1}^{t-1}\tilde{\tilde{\mathbf{v}}}_{s,\beta_2,t-1} + (1 - \beta_2)||\mathbf{v}_t||^2$, to verify $\alpha_t \ge \alpha_{t+1} \implies \sum_{s=1}^{t-1}\tilde{\tilde{\mathbf{v}}}_{s,\beta_2,t-1} \le \sum_{s=1}^{t}\tilde{\tilde{\mathbf{v}}}_{s,\beta_2,t}$, it suffices to verify

$$\Longleftarrow \sum_{s=1}^{t-1}\tilde{\tilde{\mathbf{v}}}_{s,\beta_2,t-1} \le ||\mathbf{v}_t||^2$$

$$\Longleftarrow (1 - \beta_2)\sum_{s=1}^{t-1}\beta_2^{t-1-s}||\mathbf{v}_s||^2 \le ||\mathbf{v}_t||^2$$

$$\Longleftarrow \max_{s\in[t-1]}(||\mathbf{v}_s||^2)(1 - \beta_2)\sum_{s=1}^{t-1}\beta_2^{t-1-s} \le ||\mathbf{v}_t||^2$$

$$\Longleftarrow \max_{s\in[t-1]}(||\mathbf{v}_s||^2)(1 - \beta_2^{t-1}) \le ||\mathbf{v}_t||^2$$

$$\overset{(i)}{\Longleftarrow} \max_{s\in[t-1]}(||\mathbf{v}_s||^2)\frac{1}{a} \le ||\mathbf{v}_t||^2$$

$(i)$ assumes $\beta_2 \in \left(1 - \frac{1}{a(T-1)}, 1\right)$ universally where $a > 1, T > 1$. Thus, we have $\beta_2^{t-1} = \left((1 - \frac{1}{a(T-1)})^{t-1}, 1\right)$, where its left hand side has lower bound $(1 - \frac{1}{a(T-1)})^{t-1} \ge (1 - \frac{1}{a(T-1)})^{T-1} \ge 1 - \frac{1}{a}$. It suffices to have $\beta_2^{t-1} \in \left(1 - \frac{1}{a}, 1\right) \implies (1 - \beta_2^{t-1}) \in (0, \frac{1}{a})$. Then, $a$ is some tunable parameter to satisfy $\max_{s\in[t-1]}||\mathbf{v}_s||^2 \le a||\mathbf{v}_t||^2$.

**Verifying (C.3).** $||\tilde{\mathbf{v}}_{t,\beta,T}||^2 \le (1 - \beta_2)\tilde{\tilde{\mathbf{v}}}_{t,\beta_2,T}$

To verify $||\tilde{\mathbf{v}}_{t,\beta,T}||^2 \leq (1-\beta_2)\tilde{\tilde{\mathbf{v}}}_{t,\beta_2,T}$, it suffices to verify

$$\Longleftarrow \quad (1-\beta)^2(\beta^2)^{T-t}||\mathbf{v}_t||^2 \leq (1-\beta_2)^2\beta_2^{T-t}||\mathbf{v}_t||^2$$

$$\Longleftarrow \quad \beta \geq \beta_2 \text{ and } \beta^2 \leq \beta_2$$

$$\Longleftarrow \quad \beta \in (\beta_2, \sqrt{\beta_2}).$$

**Proving (C.2).** $||\mathbf{z}_t|| \leq \eta$

$$||\mathbf{z}_t|| = \eta \frac{||\sum_{s=1}^{t-1} \tilde{\mathbf{v}}_{s,\beta,t-1}||}{\sqrt{\sum_{s=1}^{t-1} \tilde{\tilde{\mathbf{v}}}_{s,\beta_2,t-1}}}$$

$$\leq \eta \frac{\sum_{s=1}^{t-1} ||\tilde{\mathbf{v}}_{s,\beta,t-1}||}{\sqrt{\sum_{s=1}^{t-1} \tilde{\tilde{\mathbf{v}}}_{s,\beta_2,t-1}}}$$

$$\leq \eta \frac{\sqrt{t-1}\sqrt{\sum_{s=1}^{t-1} ||\tilde{\mathbf{v}}_{s,\beta,t-1}||^2}}{\sqrt{\sum_{s=1}^{t-1} \tilde{\tilde{\mathbf{v}}}_{s,\beta_2,t-1}}}$$

$$\leq \eta \frac{\sqrt{t-1}\sqrt{1-\beta_2}\sqrt{\sum_{s=1}^{t-1} \tilde{\tilde{\mathbf{v}}}_{s,\beta_2,t-1}}}{\sqrt{\sum_{s=1}^{t-1} \tilde{\tilde{\mathbf{v}}}_{s,\beta_2,t-1}}}$$

$$\leq \eta$$

where the first inequality is by Triangle inequality, the second inequality is by Cauchy-Schwarz inequality (considering the sum as the multiplication between all-ones vector and vector consisting of each element of the sum), the third inequality is by $||\tilde{\mathbf{v}}_{t,\beta,T}||^2 \leq (1-\beta_2)\tilde{\tilde{\mathbf{v}}}_{t,\beta_2,T}$ in (C.3), and the last inequality is by $\beta_2 = 1 - \frac{1}{a(t-1)}$ in (C.1).

**Verifying (C.4).** $||\tilde{\mathbf{v}}_{t,\beta,T}||^2 + \sum_{s=1}^{t-1} \tilde{\tilde{\mathbf{v}}}_{s,\beta_2,t-1} \leq \sum_{s=1}^{t} \tilde{\tilde{\mathbf{v}}}_{s,\beta_2,t}$

It suffices to verify

$$\overset{(i)}{\Longleftarrow} \quad (1-\beta_2)\tilde{\tilde{\mathbf{v}}}_{t,\beta_2,T} + \sum_{s=1}^{t-1} \tilde{\tilde{\mathbf{v}}}_{s,\beta_2,t-1} \leq (1-\beta_2)||\mathbf{v}_t||^2 + \beta_2 \sum_{s=1}^{t-1} \tilde{\tilde{\mathbf{v}}}_{s,\beta_2,t-1}$$

$$\Longleftarrow \quad (1-\beta_2) \sum_{s=1}^{t-1} \tilde{\tilde{\mathbf{v}}}_{s,\beta_2,t-1} \leq (1-\beta_2)||\mathbf{v}_t||^2 - (1-\beta_2)\tilde{\tilde{\mathbf{v}}}_{t,\beta_2,T}$$

$$\Longleftarrow \quad (1-\beta_2)^2 \sum_{s=1}^{t-1} \beta_2^{t-1-s}||\mathbf{v}_s||^2 \leq (1-\beta_2)||\mathbf{v}_t||^2 - (1-\beta_2)^2\beta_2^{T-t}||\mathbf{v}_t||^2$$

$$\Longleftarrow \quad (1-\beta_2)(1-\beta_2^{t-1}) \max_{s\in[t-1]} ||\mathbf{v}_s||^2 \leq (1-\beta_2)\beta_2||\mathbf{v}_t||^2$$

$$\Longleftarrow \quad \frac{1}{a} \max_{s\in[t-1]} ||\mathbf{v}_s||^2 \leq (1-\frac{1}{at})||\mathbf{v}_t||^2$$

$$\Longleftarrow \quad \max_{s\in[t-1]} ||\mathbf{v}_s||^2 \leq (a-1)||\mathbf{v}_t||^2$$

where $(i)$ is by (C.3). Then, $a$ is some tunable parameter satisfying $\max_{s\in[t-1]} ||\mathbf{v}_s||^2 \leq (a-1)||\mathbf{v}_t||^2$.

We summarize the settings of hyper-parameters $\beta_2 \in \left(1 - \frac{1}{a(T-1)}, 1\right)$ and $\beta \in (\beta_2, \sqrt{\beta_2})$. Meanwhile, $a$ is some tunable parameter satisfying $\max_{s\in[t-1]} ||\mathbf{v}_s||^2 \leq (a-1)||\mathbf{v}_t||^2$ and $a > 1$. $\square$

**Theorem 4.2** (Discounted regret of clip-free FTRL). *Using the same notations and hyper-parameter selection of Lemma 4.1, for all $T > 0$, loss sequence $\tilde{\mathbf{v}}_{1,\beta,T}, \cdots, \tilde{\mathbf{v}}_{T,\beta,T}$, comparator $\mathbf{u} \in \mathcal{D}$, i.e., $||\mathbf{u}|| \leq D$ (Assumption 2.2). Clip-free FTRL guarantees the $\beta$-discounted regret bound of*
$Regret_t^{[\beta]}(\mathbf{u}) \leq \frac{3D\sqrt{1-\beta_2}}{1-\beta}\sqrt{\sum_{t=1}^{T} \beta_2^{T-t}||\mathbf{v}_t||^2}.$

*Proof.* Firstly, we define $F_t(\mathbf{z}) := \frac{1}{2\alpha_t}||\mathbf{z}||^2 + (1 - \beta)\sum_{s=1}^{t-1}\langle\beta^{t-1-s}\mathbf{v}_s, \mathbf{z}\rangle$, thus, $\mathbf{z}_t = -\frac{\eta(1-\beta)\sum_{s=1}^{t-1}\beta^{t-1-s}\mathbf{v}_s}{\sqrt{(1-\beta_2)\sum_{s=1}^{t-1}\beta_2^{t-1-s}||\mathbf{v}_s||^2}} = \arg\min F_t(\mathbf{z})$ by setting $\alpha_t = \frac{\eta}{\sqrt{(1-\beta_2)\sum_{s=1}^{t-1}\beta_2^{t-1-s}||\mathbf{v}_s||^2}}$.

By the same notations in Lemma 4.1 and Lemma 7.1 in Orabona (2019),

$$\sum_{t=1}^{T}\langle\tilde{\mathbf{v}}_{t,\beta,T}, \mathbf{z}_t - \mathbf{u}\rangle \le \frac{1}{2\alpha_{T+1}}||\mathbf{u}||^2 + \sum_{t=1}^{T}\left[\underbrace{F_t(\mathbf{z}_t) - F_{t+1}(\mathbf{z}_{t+1}) + \langle\tilde{\mathbf{v}}_{t,\beta,T}, \mathbf{z}_t\rangle}_{\text{Component A}}\right] \qquad (2)$$

Then, the component A can be re-formulated as,

$$\begin{aligned}
&F_t(\mathbf{z}_t) - F_{t+1}(\mathbf{z}_{t+1}) + \langle\tilde{\mathbf{v}}_{t,\beta,T}, \mathbf{z}_t\rangle \\
&= F_t(\mathbf{z}_t) + \langle\tilde{\mathbf{v}}_{t,\beta,T}, \mathbf{z}_t\rangle - F_t(\mathbf{z}_{t+1}) + (F_t(\mathbf{z}_{t+1}) - F_{t+1}(\mathbf{z}_{t+1})) \\
&= F_t(\mathbf{z}_t) + \langle\tilde{\mathbf{v}}_{t,\beta,T}, \mathbf{z}_t\rangle - F_t(\mathbf{z}_{t+1}) - \langle\tilde{\mathbf{v}}_{t,\beta,T}, \mathbf{z}_{t+1}\rangle + \frac{1}{2\alpha_t}||\mathbf{z}_{t+1}||^2 - \frac{1}{2\alpha_{t+1}}||\mathbf{z}_{t+1}||^2 \\
&\le F_t(\mathbf{z}_t) + \langle\tilde{\mathbf{v}}_{t,\beta,T}, \mathbf{z}_t\rangle - F_t(\mathbf{z}_{t+1}) - \langle\tilde{\mathbf{v}}_{t,\beta,T}, \mathbf{z}_{t+1}\rangle \\
&= \underbrace{F_t(\mathbf{z}_t) + \langle\bar{\mathbf{v}}_{t,\beta,T}, \mathbf{z}_t\rangle - F_t(\mathbf{z}_{t+1}) - \langle\bar{\mathbf{v}}_{t,\beta,T}, \mathbf{z}_{t+1}\rangle}_{\text{Component A.1}} + \langle\tilde{\mathbf{v}}_{t,\beta,T} - \bar{\mathbf{v}}_{t,\beta,T}, \mathbf{z}_t - \mathbf{z}_{t+1}\rangle,
\end{aligned}$$

where the first inequality is by (C.1). in Lemma 4.1, and $\bar{\mathbf{v}}_{t,\beta,T} := \text{clip}_{\sqrt{\sum_{s=1}^{t-1}\tilde{\tilde{\mathbf{v}}}_{s,\beta_2,t-1}}}(\tilde{\mathbf{v}}_{t,\beta,T})$.

Further, the above Component A.1 can be re-formulated as

$$\begin{aligned}
F_t(\mathbf{z}_t) + \langle\bar{\mathbf{v}}_{t,\beta,T}, \mathbf{z}_t\rangle - F_t(\mathbf{z}_{t+1}) - \langle\bar{\mathbf{v}}_{t,\beta,T}, \mathbf{z}_{t+1}\rangle &\le F_t(\mathbf{z}_t) + \langle\bar{\mathbf{v}}_{t,\beta,T}, \mathbf{z}_t\rangle - \min_{\mathbf{x}}[F_t(\mathbf{x}) + \bar{\mathbf{v}}_{t,\beta,T}, \mathbf{x}] \\
&\le \frac{\alpha_t}{2}||\partial_{\mathbf{z}_t}[F_t(\mathbf{z}_t) + \langle\bar{\mathbf{v}}_{t,\beta,T}, \mathbf{z}_t\rangle]||^2 \\
&= \frac{\alpha_t}{2}||\bar{\mathbf{v}}_{t,\beta,T}||^2
\end{aligned}$$

where the second inequality is by $F_t(\mathbf{x}) + \langle\bar{\mathbf{v}}_{t,\beta,T}, \mathbf{x}\rangle$ is $\frac{1}{\alpha_t}$-strongly convex function and the property of $\mu$-strongly convex function, i.e., $f(\mathbf{x}) - f(\mathbf{x}^\star) \le \frac{1}{2\mu}||\mathbf{g}||^2$ given $f(\mathbf{x})$. The last equality is by the definition of $\mathbf{z}_t$, i.e., $\mathbf{z}_t := \arg\min F_t(\Delta)$.

Then, equation 2 can be re-formulated as

$$\begin{aligned}
\sum_{t=1}^{T}\langle\tilde{\mathbf{v}}_{t,\beta,T}, \mathbf{z}_t - \mathbf{u}\rangle &\le \frac{1}{2\alpha_{T+1}}||\mathbf{u}||^2 + \sum_{t=1}^{T}\frac{\alpha_t}{2}||\bar{\mathbf{v}}_{t,\beta,T}||^2 + \sum_{t=1}^{T}\langle\tilde{\mathbf{v}}_{t,\beta,T} - \bar{\mathbf{v}}_{t,\beta,T}, \mathbf{z}_t - \mathbf{z}_{t+1}\rangle \\
&= \underbrace{\frac{D^2}{2\eta}\sqrt{\sum_{t=1}^{T}\tilde{\tilde{\mathbf{v}}}_{t,\beta_2,T}}}_{\text{Part A}} + \underbrace{\frac{\eta}{2}\sum_{t=1}^{T}\frac{||\bar{\mathbf{v}}_{t,\beta,T}||^2}{\sqrt{\sum_{s=1}^{t-1}\tilde{\tilde{\mathbf{v}}}_{s,\beta_2,t-1}}}}_{\text{Part B}} + \underbrace{\sum_{t=1}^{T}\langle\tilde{\mathbf{v}}_{t,\beta,T} - \bar{\mathbf{v}}_{t,\beta,T}, \mathbf{z}_t - \mathbf{z}_{t+1}\rangle}_{\text{Part C}}
\end{aligned}$$

Then, we further decompose Part B and Part C.

$$\underbrace{\frac{\eta}{2}\sum_{t=1}^{T}\frac{||\bar{\mathbf{v}}_{t,\beta,T}||^2}{\sqrt{\sum_{s=1}^{t-1}\tilde{\tilde{\mathbf{v}}}_{s,\beta_2,t-1}}}}_{\text{Part B}} \le \frac{\eta}{2}\sum_{t=1}^{T}\frac{\sqrt{2}||\bar{\mathbf{v}}_{t,\beta,T}||^2}{\sqrt{||\bar{\mathbf{v}}_{t,\beta,T}||^2+\sum_{s=1}^{t-1}\tilde{\tilde{\mathbf{v}}}_{s,\beta_2,t-1}}}$$

$$\le \frac{\eta}{2}\sum_{t=1}^{T}\frac{2\sqrt{2}||\bar{\mathbf{v}}_{t,\beta,T}||^2}{\sqrt{||\bar{\mathbf{v}}_{t,\beta,T}||^2+\sum_{s=1}^{t-1}\tilde{\tilde{\mathbf{v}}}_{s,\beta_2,t-1}}+\sqrt{\sum_{s=1}^{t-1}\tilde{\tilde{\mathbf{v}}}_{s,\beta_2,t-1}}}$$

$$= \frac{\eta}{2}\sum_{t=1}^{T}2\sqrt{2}\left(\sqrt{||\bar{\mathbf{v}}_{t,\beta,T}||^2+\sum_{s=1}^{t-1}\tilde{\tilde{\mathbf{v}}}_{s,\beta_2,t-1}}-\sqrt{\sum_{s=1}^{t-1}\tilde{\tilde{\mathbf{v}}}_{s,\beta_2,t-1}}\right)$$

$$\le \sqrt{2}\eta\sum_{t=1}^{T}\left(\sqrt{\sum_{s=1}^{t}\tilde{\tilde{\mathbf{v}}}_{s,\beta_2,t}}-\sqrt{\sum_{s=1}^{t-1}\tilde{\tilde{\mathbf{v}}}_{s,\beta_2,t-1}}\right)$$

$$\le \sqrt{2}\eta\sqrt{\sum_{t=1}^{T}\tilde{\tilde{\mathbf{v}}}_{t,\beta_2,T}}$$

where

- the first inequality is by the clipping operation $\bar{\mathbf{v}}_{t,\beta,T} := \text{clip}_{\sqrt{\sum_{s=1}^{t-1}\tilde{\tilde{\mathbf{v}}}_{s,\beta_2,t-1}}}(\tilde{\mathbf{v}}_{t,\beta,T})$;

- the third inequality is by (C.4). in Lemma 4.1.

Denote $G_t = \max_{s\in[t]}\sqrt{\tilde{\tilde{\mathbf{v}}}_{s,\beta_2,t}}$ with boundary case $G_0 = 0$, then

$$\underbrace{\sum_{t=1}^{T}\langle\tilde{\mathbf{v}}_{t,\beta,T}-\bar{\mathbf{v}}_{t,\beta,T},\mathbf{z}_t-\mathbf{z}_{t+1}\rangle}_{\text{Part C}} = \sum_{t=1}^{T}\langle\tilde{\mathbf{v}}_{t,\beta,T}-\text{clip}_{\sqrt{\sum_{s=1}^{t-1}\tilde{\tilde{\mathbf{v}}}_{s,\beta_2,t-1}}}(\tilde{\mathbf{v}}_{t,\beta,T}),\mathbf{z}_t-\mathbf{z}_{t+1}\rangle$$

$$\le \sum_{t=1}^{T}||\tilde{\mathbf{v}}_{t,\beta,T}-\text{clip}_{\sqrt{\sum_{s=1}^{t-1}\tilde{\tilde{\mathbf{v}}}_{s,\beta_2,t-1}}}(\tilde{\mathbf{v}}_{t,\beta,T})||_2||\mathbf{z}_t-\mathbf{z}_{t+1}||_2$$

$$\le 2\max_{t\in[T]}||\mathbf{z}_t||\sum_{t=1}^{T}||\tilde{\mathbf{v}}_{t,\beta,T}||\left(1-\min\left(\frac{\sqrt{\sum_{s=1}^{t-1}\tilde{\tilde{\mathbf{v}}}_{s,\beta_2,t-1}}}{||\tilde{\mathbf{v}}_{t,\beta,T}||},1\right)\right)$$

$$\le 2\max_{t\in[T]}||\mathbf{z}_t||\sum_{t=1}^{T}\sqrt{\tilde{\tilde{\mathbf{v}}}_{t,\beta_2,t}}\left(1-\min\left(\frac{\sqrt{\sum_{s=1}^{t-1}\tilde{\tilde{\mathbf{v}}}_{s,\beta_2,t-1}}}{\sqrt{\tilde{\tilde{\mathbf{v}}}_{t,\beta_2,t}}},1\right)\right)$$

$$\le 2\max_{t\in[T]}||\mathbf{z}_t||\sum_{t=1}^{T}G_t\left(1-\min\left(\frac{G_{t-1}}{G_t},1\right)\right)$$

$$= 2\max_{t\in[T]}||\mathbf{z}_t||\sum_{t=1}^{T}(G_t-G_{t-1})$$

$$\le 2\max_{t\in[T]}||\mathbf{z}_t||G_T$$

$$\le 2\eta\sqrt{\sum_{t=1}^{T}\tilde{\tilde{\mathbf{v}}}_{t,\beta_2,T}},$$

where

- the third inequality is by (C.3). in Lemma 4.1, i.e., $||\bar{\mathbf{v}}_{t,\beta,T}||^2 \le (1 - \beta_2)\tilde{\tilde{\mathbf{v}}}_{t,\beta_2,t} \le \tilde{\tilde{\mathbf{v}}}_{t,\beta_2,t}$;

- the forth inequality is by $G_{t-1} \le \sqrt{\sum_{s=1}^{t-1} \tilde{\tilde{\mathbf{v}}}_{s,\beta_2,t-1}}$ and $G_t \ge \sqrt{\tilde{\tilde{\mathbf{v}}}_{t,\beta_2,t}}$;

- and the last inequality is by (C.2). in Lemma 4.1.

Then, summing over Part A, Part B, and Part C gives

$$\sum_{t=1}^{T} \langle \tilde{\mathbf{v}}_{t,\beta,T}, \mathbf{z}_t - \mathbf{u} \rangle \le \frac{D^2}{2\eta} \sqrt{\sum_{t=1}^{T} \tilde{\tilde{\mathbf{v}}}_{t,\beta_2,T}} + \sqrt{2}\eta \sqrt{\sum_{t=1}^{T} \tilde{\tilde{\mathbf{v}}}_{t,\beta_2,T}} + 2\eta \sqrt{\sum_{t=1}^{T} \tilde{\tilde{\mathbf{v}}}_{t,\beta_2,T}}$$

$$\le 3D \sqrt{\sum_{t=1}^{T} \tilde{\tilde{\mathbf{v}}}_{t,\beta_2,T}} \quad \text{(by setting } \eta = 0.38D\text{)}$$

$$\sum_{t=1}^{T} \langle \beta^{T-t} \mathbf{v}_t, \mathbf{z}_t - \mathbf{u} \rangle \le \frac{3D\sqrt{1 - \beta_2}}{1 - \beta} \sqrt{\sum_{t=1}^{T} \beta_2^{T-t} ||\mathbf{v}_t||^2},$$

which concludes the proof. $\square$

