# OpenReview forum: "Online learning meets Adam: The Road of Interpretable Adaptive Optimizer Design"
_ICLR.cc/2025/Conference — Submitted to ICLR 2025_

### Official Review · Reviewer_ecRp · 2024-11-03

**Soundness:** 2
**Presentation:** 2
**Contribution:** 2
**Rating:** 3
**Confidence:** 3

**Summary:**

The paper builds on a series of existing works to provide some theoretical analysis of Adam. Existing works have shown that Adam could be framed as an instance of the online-to-nonconvex conversion from (Cutkosky et al 2023), therefore the performance of Adam could be analyzed through the dynamic regret of the online learning algorithm it corresponds to. This paper seeks to extend this argument in two ways. First, the online learning algorithm underlying Adam is analyzed without the projection to a bounded domain. Second, the paper extends the online-to-nonconvex conversion itself towards last iteration properties.

**Strengths:**

Analyzing Adam is one of the central topics in deep learning optimization, therefore the context of this paper is relevant to the machine learning community. As far as I can see, credits are given properly to the existing works this paper builds on.

**Weaknesses:**

Although this paper raises some good points, the overall quality is in my opinion below the acceptance threshold.

First, it seems that the existing works this paper builds on already analyzed discounted FTRL without projection, such as Theorem B.2 of (Ahn et al 2024). In this regard the first contribution claimed by this paper is not new. It also seems that existing works by Ahn et al used the projected version of FTRL mainly for analytical convenience, and most of their results can be derived analogously for the version without projection, if someone is willing to do the tedious extension. So it remains unclear to me what is the new insight from the first part of the paper.

Ahn, Kwangjun, et al. "Understanding Adam optimizer via online learning of updates: Adam is FTRL in disguise." arXiv preprint arXiv:2402.01567 (2024).

The second part of the paper claims to extend the framework of Cutkosky et al to last iterate convergence, but actually the final results are not based on the last iterate, as the output still needs to be sampled from the trajectory of the iterates. Again the message here is unclear.

What's more important is that the paper claims to provide better insights on Adam, but it's unclear why the two extensions from the paper are significant in this context. I could see the paper being motivated from an analytical perspective, but for people who only want to understand why Adam is effective, what is the takeaway?

Writing needs to be thoroughly improved, as I find it hard to follow some of the arguments, such as the part of Section 4 before 4.1. There are also technicalities swept under the rug, such as the hyperparameter a in Lemma 4.1 and the associated requirements. Is there any reason we should expect those requirements to hold?

**Questions:**

Related to the above, the significance of the paper needs to be further justified, specifically for the purpose of understanding Adam.

---

> ### Author Response · Authors · 2024-11-14
>
> We thank reviewer ecRp for the valuable feedback. Below, we try to address each of the reviewer’s concerns in detail.
>
> 1. $\textit{Theorem B.2 of (Ahn et al 2024)}$ We thank the reviewer for pointing out this clarity issue. Theorem B.2 in [1] includes the term $\max_{t \in [1, T]}|\Delta_{t}|$, which is the source of the clipping operation in the following work [2]. This result indicates that the upper bound on regret depends on the maximum value of the incremental $|\Delta_{t}|$. However, rather than providing a generalized formula, a problem-dependent characteristic (value) is expected in a bound. Thus, in the following work, [2] proposes an explicit clipping strategy to achieve a clear upper bound. \
> This background motivates our own approach, where we address this issue and remove the need for such unrealistic clipping. We hope this addresses the main concern.
>
> 2. $\textit{last-iterate guarantee}$ Considering the result of last-iterate convergence of SGD for convex problem in [1], it shows that $\mathbb{E}[f(x_{T}) - f(x^{\star})] \leq \text{Constant}$ (also the high probability version: $f(x_{T}) - f(x^{\star}) \leq \text{Constant}$ with some probability). Our statement on the last-iterate guarantee offers the insight that the last iterate has a higher likelihood of being selected compared to other iterations, distinguishing it from the last-iterate guarantee statements in [3,4].
> We have clarified this distinction in the revised paper on line 452. We believe this approach aligns with practical practice, where the final few iterations are typically prioritized and guaranteed. And, it improves from the previous result in [3] that spreads convergence evenly across all iterations, and offers useful insights for future research in this area.
>
> 3. $\textit{insights on Adam}$ We thank the reviewer for pointing out this clarity issue. This work emphasizes a more interpretable design principle for Adam, refining prior results by eliminating the unrealistic clipping operation through initially analyzing the involvement of $\beta_{2}$ and offering the insight that the last iterate has a higher likelihood of being selected compared to other iterations, which also improves from previous work.\
> We acknowledge that a critical question remains: “Why is Adam effective?”. Prior research, as noted, suggests that “SGD achieves the minimax optimal convergence rate under smooth nonconvex conditions,” yet Adam often outperforms SGD in practice. The effectiveness of Adam over other methods is another emerging topic in analyzing Adam, however, this work initially focuses on interpretable Adam design and does not provide such insights.
> We do hold great interest in expanding on this topic in future work.
>
> 4. $\textit{hyperparameter}$  As noted in the discussion, the practical values typically used ($\beta, \beta_{2}$) in applications do not conform to the theoretical conditions, a limitation also observed in prior work [1]. This presents an interesting avenue for future investigation.
>
> [1] Ahn, Kwangjun, et al. "Understanding Adam optimizer via online learning of updates: Adam is FTRL in disguise." arXiv preprint arXiv:2402.01567 (2024).
>
> [2] Kwangjun Ahn and Ashok Cutkosky. Adam with model exponential moving average is effective for nonconvex optimization. arXiv preprint arXiv:2405.18199, 2024.
>
> [3] Liu, Zijian, and Zhengyuan Zhou. "Revisiting the last-iterate convergence of stochastic gradient methods." arXiv preprint arXiv:2312.08531 (2023).
>
> [4] Xiaoyu Li, Mingrui Liu, and Francesco Orabona. On the last iterate convergence of momentum methods. In International Conference on Algorithmic Learning Theory, pp. 699–717. PMLR, 2022.

---

> > ### Author Response · Authors · 2024-11-26
> >
> > Dear Reviewer, We sincerely appreciate your effort to review our paper and provide valuable feedback. As the discussion period is limited, we kindly request that you consider the new information and clarifications provided in our rebuttal. We are committed to improving our paper and addressing the concerns raised. If there are any remaining issues or aspects that require further clarification, we would be more than happy to provide additional explanations or adjustments.

---

> > > ### Comment · Reviewer_ecRp · 2024-11-27
> > >
> > > Thank you for your rebuttal. I appreciate the technical refinements in this paper compared to previous works, but they don't seem to come with genuinely novel ideas, particularly for the purpose of understanding Adam. I have to retain my score as well.

---

### Official Review · Reviewer_JZLb · 2024-11-03

**Soundness:** 3
**Presentation:** 2
**Contribution:** 2
**Rating:** 6
**Confidence:** 2

**Summary:**

The paper studies the important problem of understanding ADAM's convergence for non-convex optimization problems via online learning algorithms. Specifically, the paper considers the $\beta$-FTRL which has been shown to correspond to a version of ADAM. The main technical contributions are:

1. Removing the gradient clipping present in prior works to obtain an algorithm closer to the practically implemented version of ADAM.
2. Obtaining last iterate convergence guarantees.

I want to note that I have very limited knowledge in the field of online learning.

**Strengths:**

The paper removes the clipping used in priors works on $\beta$-FTRL, which makes the algorithm more realistic and closer to the real ADAM algorithm.

The paper obtains last iterate convergence guarantees of the order $O(\frac{1}{\sqrt{T}})$ for a general class of non-convex optimization problems.

**Weaknesses:**

[1] This work does not survey prior works' results thoroughly -- i.e, it does not state the assumptions and convergence rates obtained in prior works on ADAM.

[2] In Assumption 2.1, the bounded gradient assumption along with Assumption 2.2 of bounded domain seem very stringent. Can these be relaxed?

[3] The paper introduces the FTRL framework but does not introduce the full ADAM algorithm in detail. It would be helpful to show how it differs from the Clip-free FTRL version introduced in this paper.

Minor Comments:

In page 5, the parameter $\beta_2$ can be introduced in a better way. I had to read it off of the algorithm given below the paragraph referencing $\beta_2$, which made it very confusing.

Line 272 states "$\mathbf{z}_t$ becomes independent of $T$ and $\beta$ when $\beta$ and $\beta_2$ are appropriately chosen". This is imprecise since Lemma 4.1 shows that $\|\mathbf{z}_t\|$ is independent of these quantities.

**Questions:**

See weaknesses

---

> ### Author Response · Authors · 2024-11-18
>
> We thank the reviewer JZLb for the positive feedback. Below we try to address the questions of the reviewer.
> 1. $\textit{Assumptions in this work and results of prior work}$ In our study, particularly in the fourth point of Assumption 2.1, the bounded variance assumption for the stochastic gradient, combined with the G-Lipschitz assumption, implies a bounded gradient. Assumption 2.2 further constrains the power of the comparator $u$. These assumptions align with standard conditions for online-to-nonconvex conversions, as referenced in line 106. \
> Additionally, our work examines the classical Adam algorithm through an online learning framework, which introduces some differences in assumptions compared to prior research that typically assumes smooth nonconvex conditions, as mentioned in the related work section. Since we focus specifically on the online learning approach, we have not conducted an extensive survey of results under the smooth nonconvex assumption, but we plan to expand on this aspect upon acceptance.
>
> 2. $\textit{comparison with full ADAM algorithm}$ In the discussion section, the discrepancy between the Adam update and the clip-free FTRL update is noted, specifically the absence of bias correction terms in the clip-free FTRL formulation.
>
> 3. $\textit{Minor comments}$ Thank you for identifying these clarity issues. The modifications have been implemented in the revised manuscript.

---

> > ### Comment · Reviewer_JZLb · 2024-11-25
> >
> > Thank you for your response. I choose to retain my score.

---

### Official Review · Reviewer_ezs8 · 2024-11-03

**Soundness:** 1
**Presentation:** 3
**Contribution:** 3
**Rating:** 3
**Confidence:** 4

**Summary:**

This paper provides an analysis of non-smooth non-convex optimization that fixes some undesirable properties of previous work in an effort to be closer to the empirically successful adam algorithm. Specifically, some recent prior work developed a similarity between certain online learning algorithms and adam using an “online to non-convex conversion”. These algorithms made use of a strange “clipping” operation - essentially clipping the adam update to some fixed diameter $D$. Moreover, the convergence to critical points is provided only for a random iterate rather than perhaps the more desirable last iterate. This paper attempts to fix both issues.

**Strengths:**

The approach is intuitive, and the problem is interesting. I think if the weaknesses below could be addressed I would raise my score.

**Weaknesses:**

I have some concerns about the correctness of the results.

Lemma 4.1: the selection of $a$ seems to be impossible in general. Why should I expect to be able to do this? If any stochastic gradient happens to be zero (i.e. imagine your final loss is a hinge-loss and you happened to have a large margin on some example), then clearly $a$ must now be $\infty$. However, $a=\infty$ derails the analysis as it forces us to move to a non-discounted regime.

In Theorem 4.2 there is also a significant issue I think: it looks to me that the actual result should be $\sqrt{1-\beta_2}$ rather than $1-\beta_2$ in the numerator. Notice in line 867 to 870 in the proof it appears that the definition $\tilde{\tilde{v}}$ was applied incorrectly to get a $1-\beta_2$ outside the square root rather than inside.

In general, we should be expect this change even without looking at the proof for a mistake: notice that in the “natural” setting where $\beta_2=\beta^2$, the “correct” value for the FTRL regularizer would be $D/\sqrt{(1-\beta)^2\sum_{t=1}^T \beta_2^{T-t} v_t^2}$ for losses $(1-\beta)\beta^{T-t} v_t$. Since the provided regularizer is off by a factor $\sqrt{1-\beta}$, we should expect that to show up in the regret bound.

Regarding the last-iterate guarantee: I do not understand how it is a last-iterate guarantee. The theoretical results seem to be about still a randomly selected iterate (although admittedly with more weight on the last iterate). This still requires randomization over all iterates though, not what I usually think of when people say a last-iterate guarantee.


Mild stylistic comment on the proofs: there is a lot of use of $\rightarrow$ here, but I don’t know what this means. If you mean the standard "implies" $\implies$ arrow, then it is not correct since in many of these cases all the should then be pointing the other way since these arguments are often being used to prove the initial statement, not the final statement. As it is, I just completely ignored these arrows, and I recommend they be removed and/or possibly replaced with better explanation of the logic in appropriate cases.

Also, “it is equal to verify” is not proper grammar - try instead “it suffices to verify”.

**Questions:**

Can the issues in the proof or the last iterate guarantee be fixed?

---

> ### Author Response · Authors · 2024-11-14
>
> We sincerely thank reviewer ezs8 for the feedback. Below, we try to address each of the reviewer’s concerns in detail.
>
> 1. $\textit{The selection of $a$}$ We thank the reviewer for pointing out the boundary condition issue where $||v_{t}||^{2} = 0$, i.e., the squared norm of stochastic gradient after conversion, which results in an undefined value for $a$.Generally, a zero stochastic gradient aligns with the stopping criterion in standard optimization methods. However, for the online learning method discussed here, $v_{t} = 0$ is also `problematic’, as it yields a zero linear loss, i.e., $\ell_{t}(\cdot) = \langle v_{t}, \rangle = 0$. This does not provide feedback to inform the next update. Fortunately, zero loss, which does not impact the regret bound, offers some flexibility in handling boundary values.\
> To maintain consistency with the proofs of Lemma 4.1 and Theorem 4.2, we propose skipping updates when zero loss is encountered: if $v_{t} =\mathbf{0}$, we freeze the updating of index $t$, which results in omitting the zero term from subsequent summations and keeping the intermediate state at step $t+1$ identical to that at step $t$. This ensures that $\alpha_{t+1} \leq \alpha_{t}$, consistent with Lemma 4.1. Meanwhile, by removing zero-error loss terms, subsequent steps inherit the immediate state and continue updating according to the iterative rule, as required by Theorem 4.2. These modifications are now reflected in the revised paper on lines 268 and 678.
>
> 2. $\textit{$\sqrt{1 - \beta_{2}}$ instead of $1 - \beta_{2}$}$ We thank the reviewer for this correction, which has been addressed in the revised text at lines 879 and 304.
>
> 3. $\textit{last-iterate guarantee}$ Considering the result of last-iterate convergence of SGD for convex problem in [1], it shows that $\mathbb{E}[f(x_{T}) - f(x^{\star})] \leq \text{Constant}$ (also the high probability version: $f(x_{T}) - f(x^{\star}) \leq \text{Constant}$ with some probability). Our statement on the last-iterate guarantee offers the insight that the last iterate has a higher likelihood of being selected compared to other iterations, distinguishing it from the last-iterate guarantee statements in [1,2]. \
> We have clarified this distinction in the revised paper on line 452. We believe this approach aligns with practical practice, where the final few iterations are typically prioritized and guaranteed. And, it improves from previous result in [3] that spreads convergence evenly across all iterations, and offers useful insights for future research in this area.
>
> 4. $\textit{stylistic comment}$ All suggested stylistic changes have been incorporated in the revised manuscript.
>
> Once again, we thank reviewer ezs8 for the valuable feedback and welcome any further suggestions.
>
> [1] Liu, Zijian, and Zhengyuan Zhou. "Revisiting the last-iterate convergence of stochastic gradient methods." arXiv preprint arXiv:2312.08531 (2023).
>
> [2] Xiaoyu Li, Mingrui Liu, and Francesco Orabona. On the last iterate convergence of momentum methods. In International Conference on Algorithmic Learning Theory, pp. 699–717. PMLR, 2022.
>
> [3] Kwangjun Ahn and Ashok Cutkosky. Adam with model exponential moving average is effective for nonconvex optimization. arXiv preprint arXiv:2405.18199, 2024.

---

> > ### Comment · Reviewer_ezs8 · 2024-11-20
> > **reply**
> >
> > Regarding the skipping of zero updates: I don't think this really resolves the issue. What happens if there are some gradients that are very small but not zero, say $O(1/T)$?
> >
> > You could plausibly resolve this by adding some noise to the gradients or carefully tuning a "when to ignore the gradient because it's norm is too small" parameter, but this would destroy any adaptivity. Perhaps this is ok - even a "non-adaptive" clip-free FTRL might be interesting since the adaptivity of these algorithms is not very well characterized anyway.
> >
> > However, I think there is a more important confusion: I don't follow the final bound in Theorem 5.3, especially in light of the changed $\sqrt{1-\beta_2}$ factor.
> > First, this bound doesn't seem to show that the gradient "norm" goes to zero as $T\to\infty$, so it's not clear if the algorithm finds $(\lambda,\epsilon)$ critical points.
> > My impression is that with $\beta=1-1/T$, we'd need $\beta_2\ge 1-O(1/T)$ also,  which would make $\frac{(1-\beta)^2 T}{(1-\beta^T}\beta D\text{Regret}$ to not decay with $T$ (it would have decayed with the $1-\beta_2$ rather than $\sqrt{1-\beta_2}$ in the numerator of the regret bound).

---

> > > ### Author Response · Authors · 2024-11-22
> > >
> > > We thank reviewer ezs8 for the additional comments and valuable insights!
> > >
> > > Theorem 5.3 represents a relatively independent result, incorporating the previous method $\beta$-FTRL. However, we acknowledge the validity of your concern regarding the fixed error when applying the proposed method under the current conditions.\
> > > A more refined selection of hyperparameters, such as $\beta$, $\beta_{2}$, and $D$, has the potential to address this issue. For instance, choosing $\beta$ values closer to 1 may help mitigate the observed fixed error. Additionally, as discussed in the manuscript, bounding the increments introduces larger errors and tighter restrictions on the selection range of $\beta$. However, this trade-off also opens up possibilities for achieving more relaxed conditions under additional assumptions.

---

> > > > ### Author Response · Authors · 2024-11-26
> > > >
> > > > Dear Reviewer, We sincerely appreciate your effort to review our paper and provide valuable feedback. As the discussion period is limited, we kindly request that you consider the new information and clarifications provided in our rebuttal. If there are any remaining issues or aspects that require further clarification, we would be more than happy to provide additional explanations or adjustments.

---

### Official Review · Reviewer_GbeF · 2024-11-07

**Soundness:** 2
**Presentation:** 2
**Contribution:** 3
**Rating:** 5
**Confidence:** 3

**Summary:**

This paper studies how to interoperate the Adam algorithm with discounted online learning algorithms. The authors propose an online learning algorithm that does not need the projection operation, which aligning more closely with Adam’s practical implementation. They also provide last iterate convergence guarantees for the $\beta$-FTRL algorithm, but with a non-diminishing error.

**Strengths:**

1. This paper studies the online learning interpretation of the Adam algorithm, which is a interesting topic. Previous interpretation (Ahn & Cutkosky, 2024) requires the clipping (projection) operation, where this paper propose to avoid this operation via a careful configurations of $\beta$, which looks like a novel contribution.

2. The authors also provide a last iterate convergence guarantee the $\beta$-FTRL alorithm, which is a interesting result.

**Weaknesses:**

I have the following **major** doubts/questions:

In terms of $\beta_2$ and $\beta$: In the proof, it seems that $\beta_2$ and $\beta$ are considered as a fixed constant. However, in the Key Lemma (4.1), both $\beta_2$ and $\beta$ are time variant parameters. It leads to the following questions about the parameter $a$:  1) it seems that $a$ depend on the algorithm it self (line 292). Therefore, it is unclear to me how to find a universal $a$ to make sure such an inequality holds. It we use different $a$ at different rounds, can the proof still hold? will $\beta$ be monotone? what is the scale of a and beta?

Line 698: $1-\beta_2^{t-1}=1/(a(t-1))$: **why is it true?** we know $\beta_2=(1-1/(a(t-1)))^{t-1}$, so $1/(a(t-1))$ should equal to $1-\beta^{1/(t-1)}_2$.  $1-\beta^{1/(t-1)}_2$ and $1-\beta_2^{t-1}$ are totally different.

Since the setting of $\beta$ is critical for achieving clip-free, I believe understanding the points mentioned above are very important to make sure the proof is rigorous.

Question on the theoretical guarantees: In Theorem 4.2, what is the dependance on $\eta$? Does $v_t$ has an upper bound?

Other comments/questions:

Presentation: As mentioned above, I think the parameter setting is very unclear to me, and I hope the authors could provide more explanations.

**Questions:**

Please see the Weakness section.

---

> ### Author Response · Authors · 2024-11-14
>
> We sincerely thank reviewer GbeF for the feedback. Below, we try to address each of the reviewer’s concerns in detail.
>
> 1. $\textit{$\beta_{2}$ as a constant}$ We thank the reviewer for pointing out this issue. In the revised manuscript, we specify that $\beta_{2}$ belongs to a specific range rather than being time-variant. This change, reflected in line 696 of the revised manuscript, adjusts our findings, particularly in the setting of $\beta_{2}$, where its magnitude is now $\mathcal{O}(1 - \frac{1}{T})$, compared to the previous time-variant $\mathcal{O}(1 - \frac{1}{t})$. We hope this addresses the main concern raised.
>
> 2. $\textit{correction at line 698}$ We thank the reviewer for this correction, which has been addressed in the revised text at line 696.
>
> 3. $\textit{denpendance on $\eta$}$ The parameter $\eta$ is tunable and typically selected manually. Consistent with prior work, the optimal value of $\eta$ depends on specific problem characteristics. For our study, we set $\eta = 0.38D$ to achieve the regret bound presented in Theorem 4.2, as outlined in line 876.
>
> 4. $\textit{upper bound of $v_{t}$}$ In the conversion of the online learning method to an optimization algorithm, the environment feedback $v_{t}$ represents the stochastic gradient. In our work, particularly in the fourth point of Assumption 2.1, the bounded variance assumption for the stochastic gradient combined with the G-Lipschitz assumption does imply an upper bound. It is worth mentioning that Assumption 2.1 contains the standard conditions for non-convex and non-smooth optimization, as referenced in line 106.
>
> Once again, we thank reviewer GbeF for the valuable comments and welcome any additional suggestions.

---

> > ### Author Response · Authors · 2024-11-26
> >
> > Dear Reviewer,
> > We sincerely appreciate your effort to review our paper and provide valuable feedback. As the discussion period is limited, we kindly request that you consider the new information and clarifications provided in our rebuttal.
> > We are committed to improving our paper and addressing the concerns raised. If there are any remaining issues or aspects that require further clarification, we would be more than happy to provide additional explanations or adjustments.

---

### Meta-Review · Area_Chair_nMc4 · 2024-12-10

**Metareview:**

This paper explores the theoretical foundations of the Adam optimizer, particularly focusing on its relationship with online learning algorithms. The authors propose a clip-free FTRL method that removes the gradient clipping present in previous works, offering a theoretical framework that aligns more closely with the practical implementation of Adam.

Reviewers raised significant concerns regarding the validity of the proofs and the novelty of the contributions. Several reviewers questioned the treatment of parameters, as well as the assumptions made, particularly those related to bounded gradients. Additionally, the last-iterate convergence claims were disputed, with reviewers pointing out that the results still rely on sampling from the entire trajectory rather than focusing solely on the last iterate.

**Additional Comments On Reviewer Discussion:**

The authors have partially addressed the reviewers' concerns. However, due to significant issues in the theoretical analysis, I believe the paper requires a full review after substantial revisions.

---

### Decision · Program_Chairs · 2025-01-22

Reject